# Dietary Supplements during COVID-19 Outbreak. Results of Google Trends Analysis Supported by PLifeCOVID-19 Online Studies

**DOI:** 10.3390/nu13010054

**Published:** 2020-12-27

**Authors:** Jadwiga Hamulka, Marta Jeruszka-Bielak, Magdalena Górnicka, Małgorzata E. Drywień, Monika A. Zielinska-Pukos

**Affiliations:** Institute of Human Nutrition Sciences, Warsaw University of Life Sciences (SGGW-WULS), 159C Nowoursynowska Street, 02-787 Warsaw, Poland; jadwiga_hamulka@sggw.edu.pl (J.H.); marta_jeruszka_bielak@sggw.edu.pl (M.J.-B.); magdalena_gornicka@sggw.edu.pl (M.G.); malgorzata_drywien@sggw.edu.pl (M.E.D.)

**Keywords:** dietary supplements, immunity, COVID-19, Google Trends, internet, nutrients, vitamin D, vitamin C, zinc, fatty acids, bioactive compounds

## Abstract

The use of dietary supplements (DSs) has been steadily increasing all over the world and additionally, the sales of DSs have dynamical increased in the wake of coronavirus disease 2019 (COVID-19) in most of the countries. We investigated DSs phenomenon in 2020 through (1) exploration of Google searches worldwide and in Poland (with Google Trends (GT) tool), and (2) analyses of results of PLifeCOVID-19 Online Studies conducted during the first and second wave of the pandemic. The conducted GT analysis and cross-sectional studies revealed that during the COVID-19 outbreak in March 2020, the interest in immune-related compounds and foods like vitamins C and D, zinc, omega-3, garlic, ginger, or turmeric, as well as their consumption increased. Improving immunity was the main reason behind the supplementation and changes in consumption of pro-healthy foods. GT analysis has shown these interests were positively correlated with the interest in COVID-19, but adversely with cumulative cases or deaths. Respondents tended to start supplementation during the first COVID-19 wave rather than the second one. Except for the role of vitamins D and C, zinc, and selenium in patients with deficiencies of those nutrients, there are no clear and convincing studies that support the role of DSs use in COVID-19 prevention and treatment in healthy, well-nourished individuals. Moreover, as the risk of elevated intake of some nutrients due to the popularity of DSs exists, effective education of consumers in rationale use of DSs and health-protecting behaviors against COVID-19 should be developed.

## 1. Introduction

Dietary supplements (DSs) contain one or more dietary compounds as vitamins, minerals, amino acids, or other substances with a nutritional or physiological effect. The use of DSs has been steadily increasing all over the world for the last decades, and approximately 50–75% of populations have taken them routinely, and almost half of them—regularly [1,2,3,4]. Consumers use DSs for a broad range of reasons, which depends on their age, sex, physical activity, or health status, but mainly the overall health and wellness, illness prevention, and correction of the dietary deficiencies are indicated [3,5,6]. Disturbing is that DSs are consumed by people without any clinical signs or symptoms of deficiency, and their effectiveness in such conditions is questioned. However, it should be remembered that dietary supplements should not replace a healthy and balanced diet, but correct the deficiency or maintain an adequate intake of a certain nutrient [7].

Recently, due to the spread of the coronavirus disease 2019 (COVID-19) pandemic in 2020, a new reason for DSs use has emerged. By December 2020, over 25,000 scientific publications on COVID-19 or severe acute respiratory syndrome coronavirus 2 (SARS-CoV-2) had already been published [8]. At the same time, not only scientists, but also consumers have shown a high interest in information related to COVID-19. In media, more and more advertisements of dietary supplements that are thought to help in the treatment and prevention of COVID-19 can be found [9,10]. Although, the scientific evidence on immune-boosting, anti-inflammatory, antioxidant, and antiviral properties of several bioactive compounds exists, actually the guidelines for the treatment of COVID-19 do not comment on the use of DSs [9,11]. 

The global dietary supplement market was valued at around USD 101.38 billion in 2018 and was expected to doubled that amount in 2020 (approx. USD 220.3 billion), with the further grow [6]. Considering global and national data on the DSs market before pandemic, it was prognosed to rise by around 7% annually through 2025 generally in the world, and 5% in Poland [12]. However, according to a report published in October 2020, the sales of dietary supplements had increased dynamically in the wake of COVID-19 in most of the countries, also in Poland. At the beginning of the pandemic, some types of DSs recorded even triple-digit growth rates [13]. 

The Internet plays a key role in obtaining information, as it provides instant access to many sources, and enables interaction with other users [14,15]. Google Trends is one of the most common available tools to track such interest. Analyses of searches related to COVID-19 and dietary supplements can reveal which dietary compounds are of the greatest interest and how that interest fluctuates over time. So far, only a few studies have looked at Google queries about dietary supplements, including that during the SARS-CoV-2 pandemic [16,17]. Therefore, in this paper, we investigated dietary supplements phenomenon in 2020 through (1) exploration of Google searches worldwide and in Poland, and (2) analyses of results of PLifeCOVID-19 Online Studies conducted during the first and second wave of the pandemic. Firstly, we used Google Trends (GT) (1) to analyze changes in searching of DSs and immune-related topics since the beginning of 2020; (2) to investigate the association between searching of those topics and COVID-19; (3) to explore geographical distribution of searching those terms; (4) to compare popularity of DSs-related searches since the beginning of 2020. Secondly, we analyzed data about the use of DSs during the COVID-19 pandemic from two cross-sectional studies collected in April–May 2020 (PLifeCOVID-19 Online Study 1) and November 2020 (PLifeCOVID-19 Online Study 2).

## 2. Materials and Methods 

### 2.1. Confirmed COVID-19 Cases Data Acquisition 

We collected data about daily cumulative COVID-19 confirmed cases and deaths between 1 January 2020 and 31 October 2020 from collection of Our World in Data supported by researchers from the University of Oxford [18]. Based on those data, we extracted weekly cumulative COVID-19 cases and deaths globally in the world, and separately in Poland and Europe. We analyzed weekly data because data from Google Trends were given at weekly intervals.

### 2.2. Google Trends Data Acquisition 

In this paper, we used data obtained within the online tool Google Trend (GT [19]). GT allows to estimate the relative search volume (RSV) of searches made by Google users in a given period and area. RSV takes values between 0 and 100, where 0 represents complete lack of interest and 100 indicates the peak of popularity in a given area and period. In GT, it is possible to analyze a “search term” (literal typed term) or a “topic” (all phrases related to popular queries), which allows to analyze all related queries in all available languages. Additionally, GT allows to compare up to five terms at the same time [20].

For the purpose of this study, we collected GT data since 1 January 2020 up to 31 October 2020, both worldwide and separately in Poland. Those data cover the beginning of the pandemic, when the confirmed cases were reported only for the Wuhan City, Hubei province in China (first case outside China was reported on 13 January in Thailand; 21 January in North America; 24 January in Europe; 25 January in Africa, Australia; and 26 February in South America), and the first and second wave of pandemic after its outbreak [21]. In Appendix A is presented all detailed information about the conducted GT searching according to recommendations in literature [16,20]. Shortly, in the present study, we searched topics related to coronavirus, immune system, dietary supplements, nutrients, bioactive compounds, and foods related to dietary supplements and immune system. We chose those topics based on literature searches [11,14,16,17,22,23] and results obtained within the PLifeCOVID-19 Online Study which we conducted in April–May 2020 [24,25]. We searched those topics separately (non-adjusted data), as well as each topic in comparison to the topic “lutein” (adjusted data) which allowed to compare the popularity of all searched topics.

### 2.3. PLifeCOVID-19 Online Study 1 and 2

#### 2.3.1. Study Design and Participants

The two editions of the cross-sectional PLifeCOVID-19 Study were conducted in two time-frames: (1) in April and May 2020 during the first wave of the pandemic and the nationwide quarantine due to COVID-19 in Poland, and (2) in November during the second wave of the pandemic when an increase in the number of infections and deaths was observed in Poland. These cross-sectional surveys were conducted among adults living in Poland with the use of the Google Forms web survey platform. The studies were conducted in accordance with the Helsinki Declaration [26] and participation was entirely voluntary. The questionnaires were anonymous and no personal data were collected, therefore no informed written consent was requested. In the first edition, we collected 2575 responses and analyzed 2296, whereas in the second edition, we collected 1059 responses but in analysis, we included 978. Details on the inclusion and exclusion criteria are presented in Figure 1.

The first edition of PLifeCOVID-19 aimed to investigate the changes in lifestyle behaviors, including dietary ones, during the lockdown. Detailed description of the questionnaire and methods have been reported previously [24,25]. As the data about DSs usage during the pandemic were limited to three questions [24], we decided to conduct an extended survey with more detailed questions about DSs use and changes in the consumption of immune-related food products.

#### 2.3.2. PLifeCOVID-19 Online Study 2 Questionnaire

##### Dietary Supplements Data

The first part of the questionnaire consisted of questions about the use of DSs. In regard to this study, we defined them as all products that are concentrated sources of vitamins, minerals, fatty acids, probiotics, or other bioactive compounds and herbs, and allow precise dosing. The use of dietary supplements in the COVID-19 pandemic and before were addressed by the following questions: (i) Do you take any dietary supplements? If *yes*: What dietary supplements or medications containing nutrients (e.g., vitamins, minerals, fatty acids) do you take during the pandemic? Please provide the trade name of the preparation or the ingested nutrients (e.g., vitamin D, C, zinc, probiotic, ...). (ii) Do you take the same supplements as you took before the pandemic? Yes, all/Yes, but only some/No, I have started taking them during the pandemic.

In addition, we asked about the use of bioactive compounds, herbs, and pro-health foods such as ginger, garlic, elderberry, turmeric, *Nigella sativa*, etc.: Has your consumption of the following products changed in the fall of 2020? For what reason(s) have you changed your consumption of these products?

Based on data about trade names of preparations given by subjects and information gained from online shops and pharmacies, we analyzed vitamins, minerals, fatty acids, bioactive compounds, and herbs taken with DSs. 

Additionally, we asked respondents if they had made any changes in their consumption of herbs or foods traditionally related to the immune system during the second wave of pandemic, as well as the reasons for their decisions.

##### Other Data

The second part of the questionnaire included questions regarding the general characteristics of respondents, such as age, gender, body mass and height, education level, housing situation, place of living, and employment forms during pandemic. Those parts were analyzed according to the methods described previously by Górnicka et al. [24].

### 2.4. Statistical Analysis

We conducted all statistical analysis in Statistica 13.3 (TIBCO Software Inc., Paolo Alto, CA, USA) and Excel for Microsoft 365 (Microsoft, Redmond, WA, USA). The *p*-value below 0.05 was considered as significant in all conducted analysis.

#### 2.4.1. Analysis of GT Data

We use non-adjusted RSVs (relative search values)data for analysis of their changes in the study period, association with COVID-19, and geographical distribution of searches. After checking the normality of distribution with the Kolmogorov–Smirnov test, we used the Spearman’s rank correlation to explore the relationship between non-adjusted RSVs of dietary supplements-related topics and COVID-19 confirmed cumulative cases and deaths and coronavirus’ RSV. Those analysis were conducted separately for global data (global RSVs and global COVID-19 data) and Polish data (Polish RSVs and global, European, and Polish COVID-19 data) between 1 January and 31 October 2020. 

We used adjusted RSVs data for the comparison of the popularity of searched topics within an analyzed timeframe both globally and separately in Poland. We calculated RSVs of all topics in proportion to the topic “lutein” which was set as 1.0.

#### 2.4.2. Analysis of PLifeCOVID-19 Online Study 1 and 2

Datasets from PLifeCOVID-19 Studies 1 and 2 were analyzed separately. All variables were analyzed qualitatively and reported as percentages (%) and numbers (*n*) according to Cole [27] recommendation. Socio-demographic data were compared among the three subgroups according to the use of DSs: (1) supplementation started before the COVID-19 pandemic; (2) supplementation started during the COVID-19 pandemic; (3) lack of supplementation. Differences among groups were analyzed using the chi-square test.

## 3. Results

### 3.1. GT Analysis of Immune-Related Nutrients, Bioactive Compounds, and Herbs

Globally, in relation to the coronavirus, the following were searched: vitamins; vitamin D; vitamin K; vitamin C; zinc; selenium; garlic; onion; echinacea; lactoferrin; elderberry; *Nigella sativa*; and *Glycyrrhiza glabra*, whereas in Poland the searches included only vitamins; vitamin D; vitamin C; and *Glycyrrhiza glabra* (Appendix A). Searches in relation to the antiviral properties were conducted for iodine, ginger, and turmeric (worldwide), whereas in Poland—for garlic, onion, and ginger. In Poland, fish oil was searched in relation to immunity.

#### 3.1.1. Time and Geographical Distribution of RSVs 

In Figure 2A–D and Appendix A, we present RSVs for nutrients, bioactive compounds, and herbs popular in DSs, as well as for queries coronavirus and dietary supplement in relation to the confirmed cumulative COVID-19 cases in the world. At the beginning of the pandemic, when the first wave of COVID-19 spread all around the world, we observed the highest peak in RSVs of coronavirus, immune system, vitamin C, zinc, selenium, garlic, ginger, turmeric, honey, echinacea, elderberry, *Nigella sativa*, and *Glycyrrhiza glabra*. In the contrary, the highest peaks during the second wave of COVID-19 were observed for vitamins, vitamin D, and lactoferrin, whereas the increase in the interest in this period was observed for vitamins C and K, zinc, selenium, acerola, *Nigella sativa*, and *Glycyrrhiza glabra*. In Poland (Appendix A), we observed similar trends for coronavirus, immune system, vitamins C and D, zinc, selenium, garlic, acerola, echinacea, *Glycyrrhiza glabra*. Unlike the worldwide data, ginger, turmeric, and honey reached the highest peak in the autumn period. 

Using adjusted data, we compared RSVs in relation to lutein, which enables to compare the popularity of selected nutrients, bioactive compounds, and herbs over the analyzed period in the world (Appendix A), and separately in Poland (Appendix A). The five most popular compounds were: magnesium, vitamin D, iron, honey, and vitamin B_12_ (globally); magnesium, vitamin D, honey, vitamin C, and iron (Poland).

We also checked geographical distribution of analyzed searches (Appendix A). European countries tended to search: vitamins K and C, selenium, rutin, lactoferrin, elderberry, sea buckthorn, and *Glycyrrhiza glabra*. For Middle Eastern countries, they were as follows: vitamins, vitamin D, zinc, onion, raspberry, *Nigella sativa*; for Asian countries, they were: iron, iodine, fish oil, probiotic, garlic, honey; for African countries: immune system, garlic, ginger, turmeric; and for Central or South American and Caribbean countries: vitamins K and C, omega-3 fatty acids, garlic, ginger, turmeric, and echinacea. 

#### 3.1.2. Correlations between Coronavirus and RSVs

Analysis of the Spearman rank’s coefficients (Table 1 and Appendix A) revealed moderate correlations between COVID-19 cumulative cases or deaths and RSVs of lactoferrin (*r* = 0.71, *p* ≤ 0.001, for both cases and deaths), elderberry (*r* = −0.77, *p* ≤ 0.001, for both cases and deaths), and turmeric (*r* = −0.74, *p* ≤ 0.001, for both cases and deaths) in the worldwide. In Poland, such correlations were found in the case of raspberry (*r* = 0.72, *p* ≤ 0.001, for both cases and deaths) and vitamin A (*r* = −0.60, *p* ≤ 0.001, for both cases and deaths). Moderate and very strong correlations were observed between worldwide coronavirus RSV and RSVs of elderberry (*r* = 0.67, *p* ≤ 0.001), turmeric (*r* = 0.69, *p* ≤ 0.001), onion (*r* = 0.70, *p* ≤ 0.001), *Nigella sativa* (*r* = 0.71, *p* ≤ 0.001), vitamin C (*r* = 0.77, *p* ≤ 0.001), ginger (*r* = 0.84, *p* ≤ 0.001), immune system (*r* = 0.87, *p* ≤ 0.001), and garlic (*r* = 0.94, *p* ≤ 0.001). In Poland, we observed moderate association between coronavirus RSV and *Glycyrrhiza glabra* (*r* = 0.70, *p* ≤ 0.001).

### 3.2. Results of PLifeCOVID-19 Online Study 1 and 2

#### 3.2.1. Use of DSs by Polish Adults

According to the PLifeCOVID-19 Online Study first and second edition, DSs were not used by 53% and 21% respondents, respectively. Most of DSs users took all supplements before the COVID-19 pandemic (34% and 45%, respectively; Figure 3). In the second edition of the study, we observed higher rates of respondents who started supplementation during the COVID-19 pandemic. According to the results of both studies, users of dietary supplements were females, younger, better educated, living with partner and/or children, residents of urban areas, did not work, worked less, or began remote work (Appendix A).

Table 2 and Appendix A show rates of supplementation of nutrients and bioactive compounds. The most common nutrients supplemented during the first and second wave of pandemic were vitamin D (38% and 67%, respectively), vitamin C (17% and 37%, respectively), and omega-3 fatty acids (15% and 35%, respectively). Those nutrients were followed by folic acid (15%) and magnesium (15%) or vitamins B (33%) and folic acid (32%), respectively in the first and second study edition. Additionally, nutrients and bioactive compounds that started to be supplemented during the first wave of the pandemic were vitamin D (6.3% and 22%, respectively for Study 1 and 2), vitamin C (4.7% and 13%, respectively), omega-3 fatty acids (2.8% and 8.2%, respectively), zinc (2.7% and 12%, respectively), and vitamins E and A (2.6%) in Study 1 and vitamins B (9.4%) in Study 2. Moreover, the study conducted during the second wave (PLifeCOVID-19 Online Study 2) revealed that supplementation was initiated during the first wave of the pandemic more often than in the second one.

#### 3.2.2. Reasons for the Usage of Dietary Supplements

Reasons for the usage of DSs were obtained within the second edition of the PLifeCOVID-19 Online Study and are presented in Figure 4. Improving immunity was declared most often (60%), followed by improving overall health and wellness (57%), seasonal vitamin D or fish oil use (56%), and filling the nutrients’ gaps in the diet (53%). Respondents who started supplementation during the pandemic did that because they wanted to improve the immunity and/or to be protected against COVID-19 (13%); because of pregnancy or breastfeeding (5.0%); to improve their skin, hair, and nails (2.2%); or due to seasonal use of vitamin D or fish oil (2.0%).

The majority of respondents used supplements based on their own initiative (68%) or doctor’s, pharmacist’s, or nutritionist’s recommendation (40%; Appendix A), whereas main sources of knowledge about DSs were doctor (43%), media (40%), or books (39%; Appendix A).

#### 3.2.3. Changes of Consumption of Immune-Related Herbs and Foods

Respondents of the second edition of PLifeCOVID-19 declared most often an increase in consumption of ginger (33%), honey (33%), lemon (32%), fermented vegetables and fruits (24%), raspberry syrup (17%), garlic (17%), and turmeric (16%) (Figure 5). The main reason for higher consumption of such products was to improve immunity and/or to be protected against COVID-19, which was declared by 32% of respondents.

## 4. Discussion

The conducted GT analysis and cross-sectional PLifeCOVID-19 Studies revealed that during the COVID-19 outbreak in March 2020, the interest in immune-related compounds and foods, as well as their consumption increased. It is worth to note that improving immunity was the main reason behind the supplementation and changes in consumption of pro-healthy foods. However, as GT analysis has shown, these interests were correlated to the interest in COVID-19, but adversely to cumulative cases or deaths. Moreover, respondents tended to start supplementation during the first COVID-19 wave than the second one. 

We have observed that the worldwide interest in immune-related nutrients, bioactive compounds, and foods such as vitamins C and D, zinc, selenium, garlic, ginger, turmeric, honey, echinacea, elderberry, *Nigella sativa*, *Glycyrrhiza glabra* reached their peaks during the first or second wave of COVID-19, whereas interests in other compounds were stable over the time. Those results are partially consistent with a previous study by Mayasari et al. [17], in which increasing interest in vitamin C, zinc, garlic, ginger, onion, but not vitamins D or E, herbs, and turmeric were reported. Differences in those observations may be a result of different time-frames chosen for GT analysis (June 2019–April 2020 [17] versus January 2020–October 2020 in our study). For vitamin D, we reported the peak in an autumn season, which may be related to seasonal use of this vitamin according to guidelines and previously reported seasonal variances in its RSV [16,28], nevertheless, all top five related queries in our study were searched in the context of COVID-19. Interestingly, for onion, we noted a peak in August 2020 related to a *Salmonella* outbreak linked to onions in North America [29]. However, we observed that although four out of five top queries for onions were related to *Salmonella*, one was related to the coronavirus. 

Our population-based PLifeCOVID-19 Studies provided consistent results with GT analysis. We observed that DSs were used by 48% (Study 1) and 79% (Study 2), and DSs users were more often younger, better educated women living in more urbanized areas, which is in line with previous studies [1,3,5,30]. Interestingly, vitamins D and C were most often supplemented compounds, as also followed by zinc were most often started supplemented since the beginning of the pandemic in both editions of PLifeCOVID-19 Studies. We observed that omega-3 fatty acids were among the three most popular DSs, but were not introduced during the COVID-19 pandemic. As previous studies report, vitamin C [1,3,5], vitamin D [5], and omega-3 fatty acids [3,5] are one of the most popular nutrients taken with DSs. A recent study from Saudi Arabia found that vitamin C was used during the COVID-19 pandemic by almost all DSs users [31]. Another study from the USA reported that medication changed in barely 3% of respondents, mostly regarding introducing new vitamins or DSs, including zinc and vitamin C, since COVID-19 spread [32]. 

In the second edition of our study, we also reported that between 16 to 33% of respondents during the autumn period increased their intake of turmeric, garlic, raspberry syrup, fermented vegetables and fruits, lemon, honey, and ginger. Alyami et al. [31] also reported that around 15% of respondents used herbal products or DSs, and even twice more believed that consumption of turmeric, ginger, garlic, or onion may help to increase immunity and to reduce the risk of getting COVID-19. In our study, immunity improvement and improving the overall health and wellness were the main reasons for DSs use, declared by nearly 60% of respondents. In addition, other studies report that use of some DSs is often justified by the desire to improve immunity, overall health, to prevent illness, or to treat viral infections, even before the COVID-19 outbreak [3,5,6,7,13,31,33,34].

Indeed, there is some scientific evidence on immune-boosting, anti-inflammatory, antioxidant, and antiviral properties of several bioactive compounds and foods, including vitamins D and C, zinc, selenium, garlic, ginger, turmeric, lactoferrin, elderberry, or *Nigella sativa*, however they cannot be freely extrapolated to the effects on the SARS-CoV-2 virus [9,11,14,22,23,34]. Recently, the compounds like vitamin D, vitamin C, zinc, probiotics, curcumin, quercetin have been extensively studied in the context of coronavirus [34,35,36]. Initial surveys have found them to possess an anti-SARS-CoV-2 effect, and suggest they might provide both prophylactic and adjuvant therapy against COVID-19 effects, therefore, they are being fast-tracked into clinical trials [34,35]. Recent studies have linked deficiency of zinc [35], selenium [37], or vitamin C [36,38] to poorer COVID-19 outcomes. Existing studies have shown that adequate serum levels of vitamin D and C may reduce the incidence of a COVID-19-related “cytokine storm” [36,39], which is further correlated to lung injury and unfavorable prognosis of COVID-19 [40]. However, low vitamin D concentrations might not only cause inflammation but might result from inflammation in the organism, showing the possible reverse causality between vitamin D and inflammatory processes [41]. Several studies investigated the role of high doses of vitamin D [42,43] or vitamin C [36] in COVID-19 patients with promising evidences on their ability to reduce not only the severity of COVID-19 but also mortality. Currently, mass administration of vitamin D supplements to populations at risk for COVID-19 is commonly recommended [11,39,42,44]. Above mentioned pieces of evidence suggest that the use of DSs may be necessary to obtain and/or maintain the optimal levels of those nutrients, specifically in individuals with higher risk of their deficiencies, including the elderly, obese, or with co-morbidities, like diabetes [45]. On the other hand, results concerning the role of supplementation in the prevention of COVID-19 in healthy and well-nourished individuals are still not fully investigated [34]. Analysis of 13 guidelines regarding nutrition during the pandemic revealed that in eleven documents, the use of DSs in COVID-19 prevention was not mentioned at all, while in the remaining two, the authors remarked that DSs use was acceptable but to meet the nutrients’ recommendations [10]. Due to the fact that in developed countries, including Europe, the intake of most of the vitamins and minerals, except vitamin D, iron, and iodine in certain populations, generally meets their recommendations [46], DSs use is not entirely justified [10] and should be monitored to prevent excessive intake of nutrients [47].

Interestingly, we observed moderate and very strong positive correlations between interest in compounds and foods with immune-boosting properties (as elderberry; turmeric; onion; *Nigella sativa*; vitamin C; ginger; garlic) and coronavirus RSV, but adverse correlations with worldwide cumulative COVID-19 cases and deaths (for elderberry, turmeric). Our results are inconsistent with those presented by Mayasari et al. [17], but once again, the possible explanation might be the different time frames of GT analysis in both surveys. Our study indicated that the interest in the above mentioned terms was the highest at the beginning of the pandemic (when also a peak in coronavirus RSV was observed), but the cumulative numbers of cases and deaths were the lowest, and this interest has not increased proportionally to the development of the COVID-19 pandemic. We observed similar results in the second PLifeCOVID-19 Study, in which higher percentage of respondents declared the introduction of supplementation with immune-related nutrients and compounds during the first wave of the pandemic (and lockdown), when COVID-19 cases and deaths in Poland were incomparably lower than in the second wave during autumn. Previous GT analysis and population-based studies revealed that the interests [48,49,50] and practices [51] of protective behaviors, such as hand washing or use of hand sanitizers increased after the COVID-19 outbreak. Moreover, undertaken of those health-protective behaviors was related to a variety of factors, including sociodemographic [51,52,53], psychological characteristic (e.g., emotion regulation [54]), perception of being at risk to get COVID-19 [54], media or social-media exposure and trust in information from media and other people (friends and bloggers) [53,55,56], and knowledge or beliefs about COVID-19 [53], however there is a limited number of studies comparing both waves of the pandemic [54]. 

Although COVID-19 vaccines have already been invented, but have not been available for a broad use yet [57], it is still necessary to use effective preventive actions against COVID-19 spread, including social isolation, physical distance, washing hands, wearing masks, but DSs usage is not mentioned [9,10,58]. Regarding dietary interventions, a healthy diet abundant in vegetables, fruits, and whole grains, and concurrently restricted in salt, fat, and sugar is recommended during the COVID-19 pandemic [10]. Nevertheless, previous results showed that during the pandemic, many people worsened their diet or/and gained weight [24,25,59,60,61]. Unfortunately, knowledge about COVID-19 prevention is not fully satisfactory [31,52,62] and often derives from social-media or media [52,62]. As it was previously described, Internet information about methods of COVID-19 prevention and treatment, or immune system boosting, including the use of DSs, varies significantly depending on the type of website (e.g., governmental or commercial) and sometimes has an inadequate quality or even presents potentially harmful information quite often [14,15,50,63]. Interestingly, during lockdown, less respondents from Poland declared to have a contact with DSs advertisements, but more believed that DSs quality was well controlled compared to the pre-pandemic period [64]. However, the COVID-19 pandemic created an increased requisition for immune-boosting products, which may be used by quackery of vendors, as FDA (Food and Drug Administration) reported that above 3% of their warning letters regarded COVID-19-related drugs, DSs, or devices [65]. Future studies should examine the impact of the intake of the specific nutrients or bioactive compounds with DSs on the incidence of COVID-19 and its course in relation to nutritional status, general health, and age, and particularly, to comorbidities such as diabetes, cardiovascular diseases, hypertension, chronic lung diseases including asthma, and chronic obstructive pulmonary disease (COPD). It is necessary to more deeply investigate the role, as well as the mechanisms of action, for vitamins D and C, zinc, and selenium in patients with deficiencies of those nutrients. In addition, it should be examined whether and which DSs ingredients (vitamins, minerals, bioactive substances, herbs) show a positive effect in relation to COVID-19, caused by the SARS-CoV-2, in healthy, well-nourished individuals.

### Strengths and Limitations

This study has some strengths and limitations that should be acknowledged. To the best of our knowledge, this study is the first exploration of Google searches about immune-related nutrients, bioactive compounds and foods during both waves of the COVID-19 pandemic. In addition, it is the first study that also included population-based results regarding DSs use during the first and second wave of the COVID-19 pandemic in Poland. An important strength of this survey is the typology of the study we appropriately selected during this period and its ease of access for the participants. In fact, due to the quarantine and limitations in contacts, the online survey was an ideal research instrument, as it allowed us to recruit a large sample. Among the strengths of the online survey, we highlight the possibility of reaching the population belonging to different geographical areas and, moreover, the speed of data collection. In addition, we conducted the study in two key periods of the COVID-19 pandemic in Poland—in the first wave of the pandemic (and national lockdown) in spring and in the second wave of the pandemic in autumn, when an extensive increase in the number of infections and deaths was experienced.

Nevertheless, a weakness which needs to be highlighted was that the GT tool is not representative for all Internet searches, because it varies among countries and is most popular from several research engines [66]. Moreover, we narrowed our GT analysis to the “health” category to distinguish the searches related to the immune system from those related to e.g., culinary purposes or chemical properties. Secondly, our samples from both studies, despite differences, were characterized by relatively high proportion of younger respondents, possibly due to the dissemination manner (via social media) and familiarity with digital technologies among young people. Moreover, due to the lockdown in the first edition of the study (April—May 2020) and numerous limitations of contacts in the second edition of the study (November 2020), it was not possible to conduct face-to-face research. Another limitation was the skewness in gender and education distribution of responders, which resulted in oversampling of women, and of those with a higher (university) education level. In particular, this may be a limitation to generalize the results of our research to the entire population in Poland.

## 5. Conclusions

Our study showed the effect of COVID-19 on DSs-related behaviors. During the COVID-19 pandemic, the Interest and use of immune-related nutrients and foods, such as vitamins C and D, zinc, garlic, ginger, or turmeric increased. As there are no clear and convincing studies that support the role of dietary supplementation in COVID-19 prevention and the existing risk of elevated intake of some nutrients due to the popularity of DSs, effective education of consumers in rationale use of DSs and health-protecting behaviors against COVID-19 should be developed and introduced at local and/or national levels. 

## Figures and Tables

**Figure 1 nutrients-13-00054-f001:**
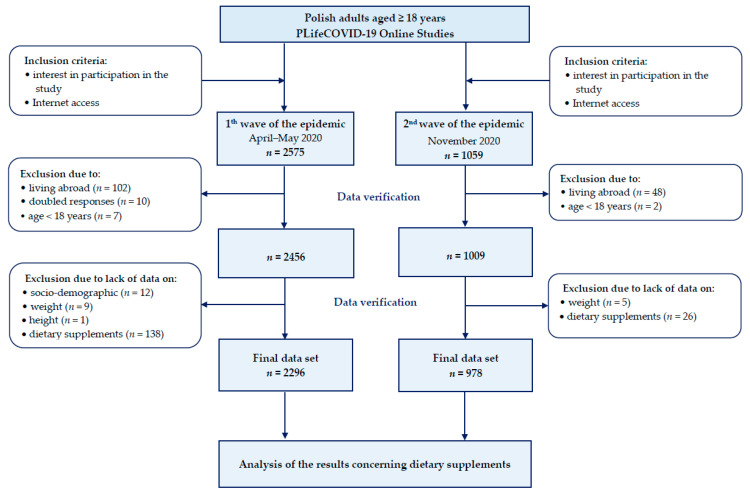
Flowchart of the creation of the final data set from participants of the PLifeCOVID-19 Study.

**Figure 2 nutrients-13-00054-f002:**
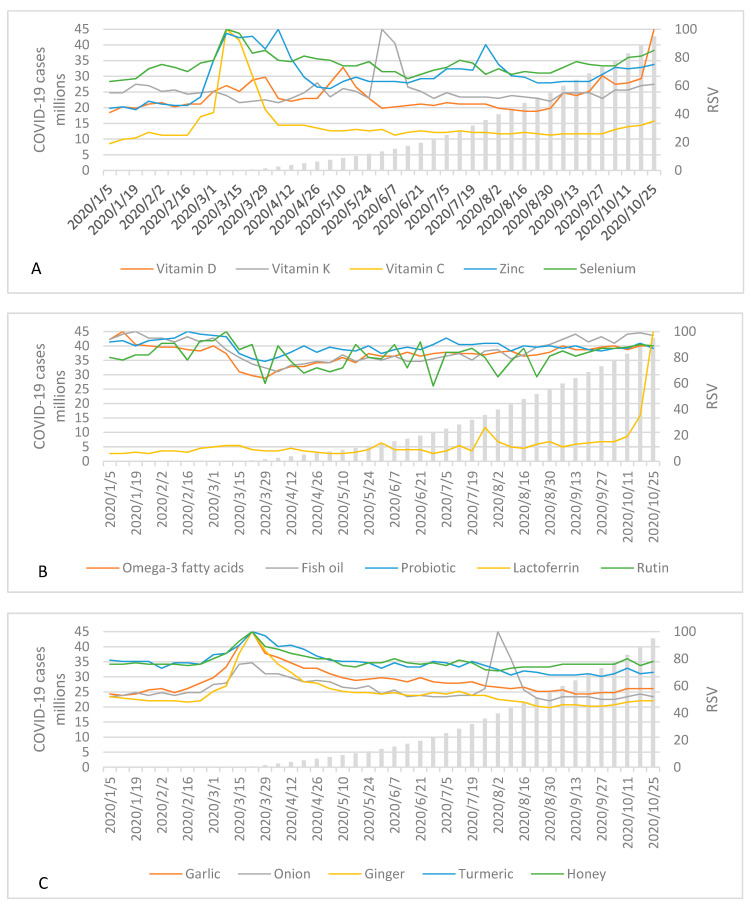
Trend curves of RSVs (relative search values) for nutrients, bioactive compounds, and herbs search queries and cumulative confirmed COVID-19 cases in the world between 1 January and 31 October2020. (**A**) vitamins D, K, C, zinc, selenium; (**B**) omega-3 fatty acids, fish oil, probiotics, lactoferrin, rutin; (**C**) garlic, onion, ginger, turmeric, honey; (**D**) echinacea, elderberry, *Nigella sativa*, *Glycyrrhiza glabra*.

**Figure 3 nutrients-13-00054-f003:**
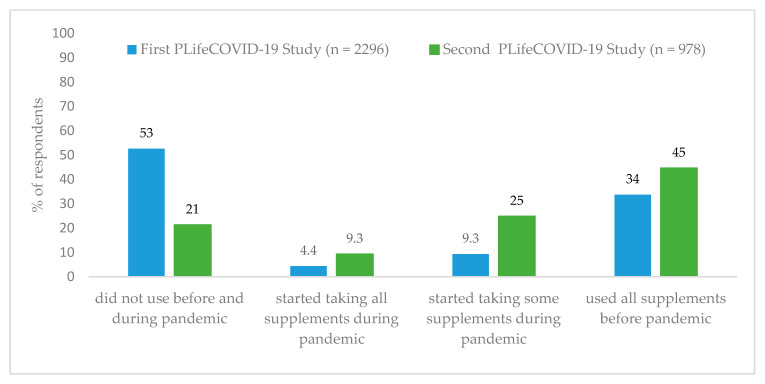
DSs (dietary supplements) usage during the first and second wave of the COVID-19 pandemic—results of the PLifeCOVID-19 Online Study 1 and 2.

**Figure 4 nutrients-13-00054-f004:**
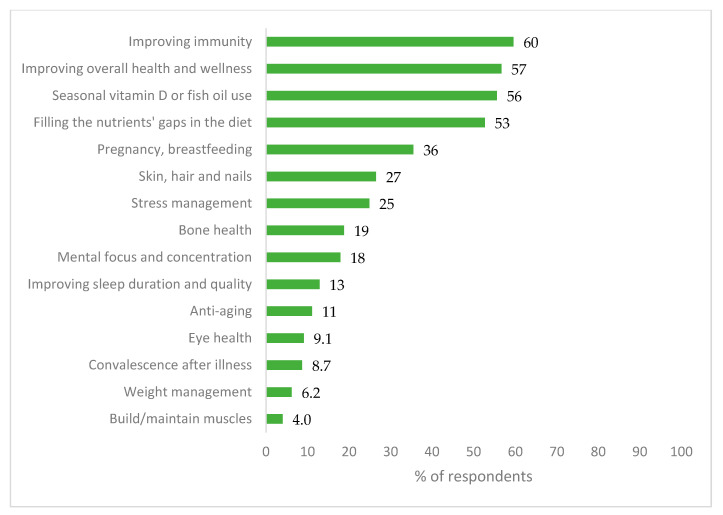
Reasons for the usage of dietary supplements—results of the PLifeCOVID-19 Online Study 2 (*n* = 978).

**Figure 5 nutrients-13-00054-f005:**
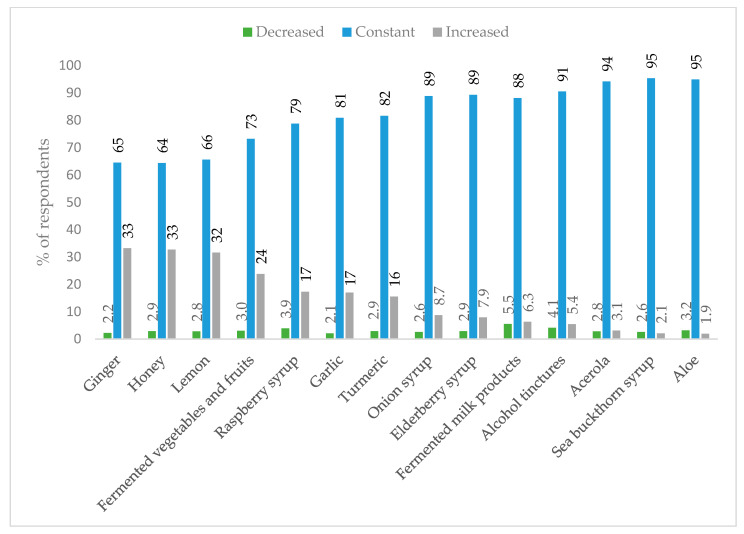
Changes in consumption of herbs and foods—results of the PLifeCOVID-19 Online Study 2 (*n* = 978) conducted during the second wave of the pandemic.

**Table 1 nutrients-13-00054-t001:** Spearman rank’s coefficients between nutrients, bioactive compounds, and herbs RSVs and cumulative confirmed COVID-19 cases, deaths, and coronavirus RSV worldwide and in Poland.

Search Query	Worldwide			Poland		
	Cumulative COVID-19 Cases	Cumulative COVID-19 Deaths	“Coronavirus” RSV	Cumulative COVID-19 Cases	Cumulative COVID-19 Deaths	“Coronavirus” RSV
Vitamin D	0.20	0.20	0.34 *	0.02	0.01	0.15
Vitamin K	−0.03	−0.03	−0.21	−0.36 *	−0.37 *	0.26
Vitamin C	0.01	0.01	0.77 ***	−0.15	−0.16	0.45 **
Zinc	0.33	0.33	0.47 **	0.03	0.02	0.52 ***
Selenium	0.05	0.05	0.57	−0.08	−0.09	0.19
Omega-3 fatty acids	0.06	0.06	−0.74 ***	0.21	0.21	−0.26
Fish oil	0.11	0.11	−0.71 ***	0.16	0.15	0.05
Probiotic	−0.26	−0.26	−0.29 ***	−0.25	−0.25	−0.12
Lactoferrin	0.71 ***	0.71 ***	−0.21	−0.11	−0.12	0.10
Rutin	−0.03	−0.03	0.03	−0.26	−0.26	0.28
Garlic	−0.25	−0.25	0.935 ***	−0.12	−0.13	0.56 ***
Onion	−0.43 **	−0.43 **	0.70 ***	−0.16	−0.16	0.32 *
Ginger	−0.48 **	−0.48 **	0.84 ***	−0.21	−0.21	0.40 **
Turmeric	−0.74 ***	−0.74 ***	0.69 ***	−0.40 **	−0.41 **	0.43 **
Echinacea	−0.09	−0.09	0.38 *	0.37 *	0.35 *	0.37 *
Honey	−0.32 *	−0.32 *	0.58 ***	0.39	0.38 *	0.00
Elderberry	−0.77 ***	−0.77 ***	0.67 ***	0.30	0.29	0.05
*Nigella sativa*	−0.08	−0.08	0.71 ***	−0.48 ***	−0.49 ***	0.04
*Glycyrrhiza glabra*	−0.43**	−0.43 **	0.21	−0.32*	−0.33	0.70 ***

RSV—relative search value; * *p* ≤ 0.05; ** *p* ≤ 0.01; *** *p* ≤ 0.001.

**Table 2 nutrients-13-00054-t002:** Nutrients and bioactive compounds supplemented during the COVID-19 pandemic—results of the PLifeCOVID-19 Online Study 1 and 2.

Nutrients	PLifeCOVID-19 Online Study 1 (*n* = 2296)		PLifeCOVID-19 Online Study 2 (*n* = 978)		
	Supplemented during Pandemic	Supplementation Started during Pandemic	Supplemented during Pandemic	Supplementation Started during First Wave of Pandemic	Supplementation Started during Second Wave of Pandemic
Vitamin D	38 (861)	6.3 (145)	67 (658)	22 (217)	4.9 (48)
Vitamin C	17 (385)	4.7 (109)	37 (363)	13 (131)	3.6 (35)
Vitamin K	3.3 (76)	0.8 (18)	7.3 (71)	1.2 (12)	0.3 (3)
Zinc	9.8 (224)	2.7 (62)	25 (244)	12 (113)	1.5 (15)
Selenium	7.2 (165)	1.1 (25)	19 (186)	6.4 (63)	1.0 (10)
Omega-3	15 (341)	2.8 (64)	35 (337)	8.2 (80)	1.0 (10)
Probiotics	3.0 (70)	0.9 (20)	6.0 (59)	2.2 (22)	0.2 (2)
Rutin	1.6 (37)	0.7 (15)	6.3 (62)	2.8 (27)	0.8 (8)
Garlic	0.3 (8)	0.04 (1)	0.9 (9)	0.3 (3)	0.1 (1)
Tumeric	0.2 (5)	0.1 (3)	2.4 (23)	1.0 (10)	0.4 (4)
Ehinacea	0.1 (2)	0.1 (2)	0.1 (1)	0 (0)	0 (0)
Ginger	0.3 (7)	0.04 (1)	0.2 (2)	0.2 (2)	0 (0)
Elderberry	0 (0)	0 (0)	0.7 (7)	0.3 (3)	0.2 (2)
*Nigella sativa*	<0.1 (1)	0 (0)	1.5 (15)	0.8 (8)	0.5 (5)
*Glycyrrhiza glabra*	0.04 (1)	0.04 (1)	0.3 (3)	0.3 (3)	0 (0)

## Data Availability

The dataset on which this paper is based is too large to be publicly archived with available resources. These data are available from the corresponding author.

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
