# Peer review of "Dietary Supplements during COVID-19 Outbreak. Results of Google Trends Analysis Supported by PLifeCOVID-19 Online Studies"

_nutrients, 2020, doi:10.3390/nu13010054_

Round 1

Reviewer 1 Report

Hamulka J and al. have performed an interesting study. They tried to compare and discuss both "facts and fads" about dietary supplements (DSs), by analysing either data from search tools, or efficacy of DSs as actually  derivable from increasingly accumulating epidemiologic data on SARS-CoVid2/Covid 19 pandemic. The only limit of this study is intrinsic to the argument: DSs have  great marketing appeal (I consider this kind of comunications mostly as fads), poor  efficacy on field (facts). By reading Hamulka J and al. paper, I was surprised in discovering some DSs predicted to have effects, like Glycirriza glabra, among others, which in some country evidently had, while in other had not peculiar commercial appeal. Another item I have appreciated, and not resolved  is that they took into account the pivotal question of efficiency of DSs linked to deficiency or sufficency of intake of some biologically indispensable compounds. For some vitamins, this is a peculiar problem. It seems quite obvious that deficiencies compromise integrity of function, and supplementation would be of help: disentangling the effects on the different populations with normal or sub-normal reserves without extensive sudies, is impossible. But, this lack of informations is not a limit of Hamulka J et al. paper or analyses, it is linked to incomplete availability of informations on the entire populations. Although the paper is worth of being published as it is (I suggest just to revise some rare spelling defect linked to automatic corrections, as on line 280, "are ... consisted" should be "consistent" or "are" should be eliminated?) I would also suggest Hamulka J and al., if they can, to evaluate and discuss, hereor in another paper how "immune system efficiency" is comunicated , and thus how some fads are promoted by advertising: "cytokine storm" may also be defined as an hyper-activation of immune response, not certainly as a blunted response, and the use of corticosteroids, which down-regulate immune response,  is one of the very few medical procedures that have been widely accepted as therapy in COVID19 most serious cases.

Author Response

Dear Reviewer,

Thank you for all your work on our manuscript “Dietary Supplements During COVID-19 outbreak. Results of Google Trends Analysis Supported by PLifeCOVID-19 Online Studies”. Your comments and suggestions were very useful and helped to improve the paper considerably. All your suggestions have been taken into account in the recent revision of the manuscript. You can find answers to your specific comments below.

Sincerely,

Monika Zielińska-Pukos

Reviewer 2 Report

This manuscript provides an interesting and useful documentation of Google searches for dietary supplements that might be useful in reducing risk of SARS-CoV-2 infection and COVID-19. While the authors did not know of publications reporting reduced risk or severity of SARS-CoV-2 or COVID-19, such publications now exist for vitamin D as discussed below. While only two are randomized controlled trials, the other studies are also valid scientific results.

“As there are no clear and convincing studies that
25 support the role of dietary supplementation in the COVID-19 prevention”

Comment: This statement is no longer correct for vitamin D as a search of pubmed.gov,  scholar.google.com and even Nutrients will show. Figure 2A shows a rapid increase in vitamin D in October, likely related to the increased publicity in Spain and the UK. Table 3 shows that vitamin D has the strongest interest of all supplements considered.

References to consider (note that some of the findings are for 25(OH)D, but with concentrations affected by vitamin D supplementation)

Vitamin D Supplementation Associated to Better Survival in Hospitalized Frail Elderly COVID-19 Patients: The GERIA-COVID Quasi-Experimental Study.

Annweiler G, Corvaisier M, Gautier J, Dubée V, Legrand E, Sacco G, Annweiler C.Nutrients. 2020 Nov 2;12(11):3377. doi: 10.3390/nu12113377.PMID: 33147894 Free PMC article.

METHODS: Seventy-seven patients consecutively hospitalized for COVID-19 in a geriatric unit were included. Intervention groups were participants regularly supplemented with vitamin D over the preceding year (Group 1), and those supplemented with vit …

Vitamin D and survival in COVID-19 patients: A quasi-experimental study.

Annweiler C, Hanotte B, Grandin de l'Eprevier C, Sabatier JM, Lafaie L, Célarier T.J Steroid Biochem Mol Biol. 2020 Nov;204:105771. doi: 10.1016/j.jsbmb.2020.105771. Epub 2020 Oct 13.PMID: 33065275 Free PMC article.

Vitamin D may be a central biological determinant of COVID-19 outcomes. ...In conclusion, bolus vitamin D3 supplementation during or just before COVID-19 was associated in frail elderly with less severe COVID-19 …

"Effect of calcifediol treatment and best available therapy versus best available therapy on intensive care unit admission and mortality among patients hospitalized for COVID-19: A pilot randomized clinical study".

Entrenas Castillo M, Entrenas Costa LM, Vaquero Barrios JM, Alcalá Díaz JF, López Miranda J, Bouillon R, Quesada Gomez JM.J Steroid Biochem Mol Biol. 2020 Oct;203:105751. doi: 10.1016/j.jsbmb.2020.105751. Epub 2020 Aug 29.PMID: 32871238 Free PMC article. Clinical Trial.

Short term, high-dose vitamin D supplementation for COVID-19 disease: a randomised, placebo-controlled, study (SHADE study).

Rastogi A, Bhansali A, Khare N, Suri V, Yaddanapudi N, Sachdeva N, Puri GD, Malhotra P.Postgrad Med J. 2020 Nov 12:postgradmedj-2020-139065. doi: 10.1136/postgradmedj-2020-139065.

SARS-CoV-2 positivity rates associated with circulating 25-hydroxyvitamin D levels.

Kaufman HW, Niles JK, Kroll MH, Bi C, Holick MF.PLoS One. 2020 Sep 17;15(9):e0239252. doi: 10.1371/journal.pone.0239252. eCollection 2020.

Association of Vitamin D Status and Other Clinical Characteristics With COVID-19 Test Results.

Meltzer DO, Best TJ, Zhang H, Vokes T, Arora V, Solway J.JAMA Netw Open. 2020 Sep 1;3(9):e2019722. doi: 10.1001/jamanetworkopen.2020.19722.

Regarding vitamin C, this review should be evaluated for citing in the manuscript.

Vitamin C-An Adjunctive Therapy for Respiratory Infection, Sepsis and COVID-19.

Holford P, Carr AC, Jovic TH, Ali SR, Whitaker IS, Marik PE, Smith AD.Nutrients. 2020 Dec 7;12(12):E3760. doi: 10.3390/nu12123760.

Reference 38, NIH re vitamin D, is not a useful reference since the authors did not do a careful or current (now) literature search and the NIH tends to disregard the beneficial effects of vitamin D since it competes with pharmaceutical drugs and vaccines, which benefit Big Pharma. The other NIH references are probably also weak.

Significant digits. The general rule is that no more non-zero digits should be given than are justified by the uncertainty of the value.

See "Too many digits: the presentation of numerical data"

https://www.ncbi.nlm.nih.gov/pmc/articles/PMC4483789/

If the uncertainty is greater than about 7%, only two non-zero digits are justified.

P values should be given to two decimal places unless the first two are 00 or the number lies between 0.045 and 0.050.

Thus, most of the values in Table 1 should have two, not three, decimal places.

Please review all numbers in abstract, text, tables, and figures and adjust accordingly.

  1. Association, W.M. World Medical Association declaration of Helsinki: Ethical principles for medical
    498 research involving human subjects. JAMA - J. Am. Med. Assoc. 2013, 310, 2191–2194.
    499

Comment: Please fix this reference in EndNote.

Author Response

(The authors gave the same response as above.)

Round 2

Reviewer 2 Report

Thanks for adding the references and discussion regarding vitamins C and D.

In the abstract

25 Except the role of vitamins D and C,

should be

25 Except for the role of vitamins D and C,

As for the numbers in Table 1, the text preceding it, and Table S4, they should be of this form: 0.34, 0.56, 0.23, etc., i.e., two decimal places. The numbers originially had three decimal places, such as 0.345; now they have one, 0.3, etc.

Please redo them to have two decimal places.

Otherwise, no further comments. 

Author Response

Dear Reviewer,

Thank you for all your work on our manuscript “Dietary Supplements During COVID-19 outbreak. Results of Google Trends Analysis Supported by PLifeCOVID-19 Online Studies”. Your comments and suggestions were very useful and helped to improve the paper considerably. All your suggestions have been taken into account in the recent revision of the manuscript. You can find answers to your specific comments attached.

Sincerely, 

Monika Zielińska-Pukos
